# Evaluation of Protective Immune Responses Induced in BALB/c Mice and Goats by the *Neospora caninum* Surface SRS Proteins and Interleukin-18

**DOI:** 10.3390/ani12212952

**Published:** 2022-10-27

**Authors:** Pu Wang, Xiaocen Wang, Weirong Wang, Pengtao Gong, Nan Zhang, Renzhe Zhang, Huan Zeng, Qian Sun, Wanqing Li, Xin Li, Shuqin Cheng, Xu Zhang, Xinyi Huang, Chenyang Gao, Yadong Zheng, Jianhua Li, Xichen Zhang

**Affiliations:** 1Key Laboratory of Zoonosis Research, Ministry of Education, College of Veterinary Medicine, Jilin University, Changchun 130062, China; 2Key Laboratory of Applied Technology on Green-Eco-Healthy Animal Husbandry of Zhejiang Province, Provincial Engineering Research Center for Animal Health Diagnostics & Advanced Technology, Zhejiang International Science and Technology Cooperation Base for Veterinary Medicine and Health Management, China Australia Joint Laboratory for Animal Health Big Data Analytics, College of Animal Science and Technology & College of Veterinary Medicine, Zhejiang A & F University, Hangzhou 311300, China

**Keywords:** *Neospora caninum*, IL-18, SRS17, SRS2, SRS52, immunization

## Abstract

**Simple Summary:**

*Neospora caninum* is an apicomplexan parasite with worldwide distribution, infecting cattle, goats and other kinds of warm-blooded animals. However, specific medicines for the prevention or treatment of *Neospora caninum* are still limited. This study is aimed at selecting a potential immunoprotective antigen of *N. caninum* and evaluating the adjuvant function of interleukin (IL)-18 in mice or goats; NcSRS2, NcSRS17 and NcSRS52 from the SRS superfamily were considered to be targets. The results reveal that the recombinant proteins NcSRS2, NcSRS17 and NcSRS52 can all induce humoral and cellular immune responses, and NcSRS17 showed itself to be a promising new antigen candidate. IL-18 had a certain immunoenhancing effect in both experimental animal mice and intermediate host goats and could be used as an adjuvant.

**Abstract:**

Neosporosis is caused by *Neospora caninum* (*N. caninum*), which mainly infects cattle and goats and severely threatens the animal industry. In this study, the inhibitory effects of polyclonal antiserum anti-NcSRS17, NcSRS2 and NcSRS52 were explored. Cytokines in mice or goat serum were detected after immunization. After infection, the survival of mice was recorded. The pathological changes and parasite loads were observed and detected in tissues. The results showed that anti-NcSRS2, NcSRS17 and NcSRS52 antibodies all inhibit the invasion and proliferation of *N. caninum*. The IFN-γ level in the NcSRS17 group was higher than that in the NcSRS2 and NcSRS52 groups, and higher in the NcSRS2-mIL-18 group than in the NcSRS2 group. The survival rates of mice were 16% in the positive control group, 67% in the SRS52 group, 83% in the SRS2 and mIL-18 groups and 100% in the SRS17 and SRS2-mIL-18 groups. Goats immunized with NcSRS17-gIL-18 developed high levels of IL-4, IL-12 and IFN-γ compared with those immunized with NcSRS-17. Parasite loads in the brains of animals in the NcSRS17 and NcSRS17-gIL-18 groups were significantly reduced, and were significantly lower in the NcSRS17-gIL-18 group (*p* ≤ 0.01). This study indicates that SRS17 may be an antigen candidate for vaccine development against neosporosis, and IL-18 can enhance the immune protective efficiency of antigen candidates.

## 1. Introduction

*N. caninum* is an apicomplexan protozoan parasite which mainly causes abortion and stillbirth in many kinds of animals, such as cattle and goats, and leads to neosporosis [1]. The definitive host is a canine, and the intermediate host can be a variety of warm-blooded animals, including cattle, goats, deer and so on [2]. *N. caninum* infection particularly leads to abortion or stillborn fetuses in bovines and motor nervous system diseases in newborn calves and, thus, severely damages the development of the animal industry [3]. However, no specific vaccine or drug is available to prevent or treat neosporosis. The preparation of preventative vaccines is crucial for the control of neosporosis. In regard to the development of *N. caninum* vaccines, studies on attenuated vaccines, inactivated vaccines, recombinant protein vaccines and nucleic acid vaccines have been performed, and significant progress has been made. At present, the candidate molecules for *N. caninum* vaccines include surface antigen (SAG1), surface antigen glycoprotein-related sequence (SRS), dense granule protein 7 (GRA7) and apical merozoite antigen 1 (AMA1) and have been widely studied. Surface proteins from the SRS superfamily play important roles in adhesion and invasion processes, which enables them to be possible anti-parasite targets. *N. caninum* SRS2 has been previously demonstrated to be a potential antigen [4]. Studies are attempting to obtain an understanding of other antigens from the SRS superfamily, such as NcSRS67 [5], NcSRS29B and NcSRS29C [6], but whether these antigens confer protection against *N. caninum* infection has not been investigated. The *N. caninum* vaccines based on these antigens could partly prevent neosporosis, but the protective effects still need to be improved.

Adjuvants can promote the reaction of T cells or B cells by enhancing the activity of macrophages, and participate in hapten or antigen immune response [7]. With the research on the vaccine and host immune response caused by *N. caninum*, cytokines as new molecular adjuvants can enhance the specific immune response of vaccines and have been widely studied [8]. Cytokines are a type of cell regulatory protein synthesized and secreted by immune cells and nonimmune cells after stimulation and can induce the differentiation and maturation of antigen-presenting cells, improve their antigen-presenting ability, help immune cells identify pathogens and eliminate pathogens [9]. Remarkably, interleukins (ILs) and interferons (IFNs) activate and regulate effector immune cells, which are very important in the initiation and regulation of the immune response. As a result, the cytokine adjuvants based on ILs and IFNs have been widely used in vaccines [10,11]. Recent research has shown that IL-18 combined with a recombinant protein can improve the immune effect of vaccines [10,12]. Unlike adjuvants derived from chemical and microbial sources, IL-18 has an immune-enhancing effect that can specifically and efficiently regulate the response of animals to vaccines, which involves the migration of immune cells to the inoculation site containing antigen-presenting cells (APCs) [13,14]. The DNA vaccine method is to insert the sequence encoding foreign genes into a certain expression system (such as a eukaryotic expression system), and then express the foreign genes in the host after injection, in order to make the immunized host increase protection against neosporosis [14,15,16]. IL-18 could also improve the abilities of DNA vaccines to induce humoral and cellular immune responses in cattle. A construct DNA vaccine (P1-2A-3C-pCDNA) induces relatively strong CD4+ T cell, CD8+ T cell, Th1 and Th2 cytokine responses. IL-18 is a safe, efficient and economical new immune adjuvant with good application prospects in livestock and poultry breeding [17]. As we studied previously, IL-18 showed an immune regulatory potential and can promote the Th 1 immune response against *N. caninum* infection [15]; however, whether IL-18 can be used as an adjuvant needs further study.

The live vaccine made by the tachyzoites of the attenuated *N. caninum* strain can provide a certain degree of protection for pregnant cows [16,18]. The abortion rate of pregnant cows was reduced by 10%. It can effectively promote the host’s immune response against *N. caninum*. Inactivated vaccine can induce the host to produce a certain level of humoral immunity, but its protective effect is poor [19,20]. Although the live vaccine has better immune effects than the inactivated vaccine, it has some limitations, such as potential risks of attenuated strain causing chronic infection, and the high cost of preparing a large number of cells and parasites. The recombinant vaccine has the advantages of being a safe immunogen and possessing a large immune volume and good immune effect. The mice immunized with *N. caninum* SAG1 and SRS2 recombinant protein showed that the recombinant protein can activate both Th1 and Th2 immune responses in mice and has a highly protective effect on mice [21]. However, recombinant vaccine also has the disadvantage of partial protection, and the main understandings of fundamental neosporosis pathogenesis and host immune response are still limited.

In this research, three SRS proteins, NcSRS2 (used as a positive control), NcSRS17 and NcSRS52, were selected, and IL-18 was fused with the potential vaccine antigens to evaluate their immune protective effects on *N. caninum* infection in mice or goats.

## 2. Materials and Methods

### 2.1. Cells, Parasites and Experimental Animals

MDBK and Vero cells were respectively cultured in high-glucose DMEM and RPMI 1640 medium. These two media contained 10% fetal bovine serum (Biological Industries, Kibbutz Hulda, Israel) and 10% penicillin-streptomycin solution (Biological Industries, Kibbutz Hulda, Israel). Tachyzoites from *N. caninum* (Nc-1) parasites were used for all studies, and the parasites were maintained by serial passaging in Vero cells and purified using 40% Percoll solution and centrifugation at 1500× *g* for 30 min to remove cell debris. The mixture of tachyzoites and dimethyl sulfoxide (DMSO) was stored in our laboratory at −80 °C.

Female BALB/c mice (aged 6–8 weeks) were selected from the Laboratory Animal Center of Jilin University (Changchun, China). These mice were maintained in isolated cages with sterile food and water in the animal facility of the Laboratory Animal Center of Jilin University. Goats that tested negative by ELISA for *N. caninum* were used in the present study. This study was carried out in accordance with the recommendations and guidelines from the Animal Welfare and Research Ethics Committee (license SY202007001), under protocols approved by Jilin University.

### 2.2. Recombinant Plasmid Construction

The genome of *N. caninum* was extracted from purified tachyzoites using a DNA Extraction kit (Bioer Technology, Anhui). Murine cDNA and caprine cDNA were synthetized using the PrimeScript™ 1st Strand cDNA Synthesis Kit (TaKaRa, Dalian, China) using total RNA obtained from LPS-treated murine peritoneal macrophages and caprine peripheral blood mononuclear macrophages, respectively. Murine peritoneal macrophages were isolated as previously described [21], caprine peripheral blood mononuclear macrophages were isolated by using the caprine PBMCs isolation kit (Hao Yang, Tianjin, China) according to the manufacturer’s instructions. The open reading frames of NcSRS2 (NCLIV_033250), NcSRS17 (NCLIV_068920), NcSRS52 (NCLIV_069780), mouse IL-18 (mIL-18) (NM_008360) (mature peptide) or goat IL-18 (gIL-18) (NM_001285544.1) (mature peptide) were obtained from the *N. caninum* genome, murine cDNA or caprine cDNA, respectively, by PCR amplification using synthetic primers (Table 1). The following PCR protocol was used: 5 min at 95 °C; 35 cycles of 95 °C for 30 s, 60 °C for 30 s and 72 °C for 90 s; and 10 min at 72 °C. First, primers with a linker (5′-3′ AGATCCGCCACCGCCAGAGCCACCTCCGCCTGAACCGCCTCCACC) were designed to fuse the NcSRS2 and mIL-18 genes or the NcSRS17 and gIL-18 genes. The recombinant plasmids pET-32a-NcSRS2, pET-32a-NcSRS17, pET-32a-NcSRS52, pET-32a-mIL-18, pET-32a-NcSRS2-mIL-18, pET-32a-gIL-18 and pET-32a-NcSRS17-gIL-18 were made by ligation and transformed into *Escherichia coli* strain Top10.

### 2.3. Expression and Identification of Recombinant Proteins

The above transfectants were plated on the LB solid medium containing ampicillin and cultured for 12 h, and the positive plasmids were screened. When the OD600 nm value of the transfectants was 0.5~0.6 in the LB liquid medium containing ampicillin at 37 °C, the expression of recombinant proteins was induced by IPTG (1 mM) at 16 °C overnight. The expression and solubility of recombinant proteins were analyzed by SDS-PAGE and Western blotting. The supernatant was collected and used with an affinity chromatography column (Ni Sepharose excel 5 mL) for purification. The non-target proteins with imidazole buffer containing 10 mM and 20 mM were washed in turn, and the target proteins were washed with 400 mM imidazole. The purified proteins were identified by SDS-PAGE, and then the protein concentration was detected by BCA kit (TransGen Biotech, Beijing, China), and analyzed by Western blotting. The antibodies used were a diluted anti-His-tag monoclonal antibody (TransGen Biotech, Beijing, China) (1:2000) and an HRP-conjugated goat anti-mouse IgG (1:4000) (Proteintech, Wuhan, China). The results were visualized with a chemiluminescence imaging system. The recombinant proteins were then purified with a nickel column.

### 2.4. BALB/c Mouse Immunization and Challenge

#### 2.4.1. Immunization of Mice

A total of 70 female BALB/c mice (aged 6–8 weeks) were divided into 7 groups (NcSRS2, NcSRS17, NcSRS52, mIL-18, NcSRS2-mIL-18, negative and positive; *n* = 10 per group), weighed and recorded. The mice in the negative group were not treated, and the positive group were injected with *N. caninum.* The mice in the negative and positive groups were not immunized. In contrast, the mice in the other groups were immunized by subcutaneous injection, and the mice in each of these groups were immunized three times. Among the groups, the first immunization included complete Freund’s adjuvant, and the immunization dose was 100 μg of adjuvant and immunogenic protein. The second and third immunization doses were halved to 50 μg each, and the antibody titers were determined 10 days after the third immunization. It should be noted that the immune adjuvant included in the second and third immunizations was incomplete Freund’s adjuvant, which was administered every two weeks.

#### 2.4.2. Detection of Antibodies

Blood samples were collected from the tail vein plexus before vaccination and on day 10 after vaccination. The serum was harvested and stored at −20 °C until use. Finally, the antibody titers and the ratios of IgG1 and IgG2a to total IgG were detected by indirect ELISA, and the serum was thus confirmed as the positive serum against *N. caninum* (P/N ≥ 2) [22,23].

#### 2.4.3. In Vitro Blocking Experiment with Polyclonal Antibodies

A total of 1 × 10^6^ tachyzoites were purified from the lysate of Vero cells. The tachyzoites were resuspended in RPMI-1640 and incubated with anti-SRS2, anti-SRS17 and anti-SRS52 antibodies separately at a dilution of 1:200. The complexes were then used to inoculate MDBK cells by incubation in a 12-well cell culture plate at 37 °C for 1 h. After removal of the noninvaded tachyzoites, the cells were continually cultured in the cell incubator for 30 h.

Nc-1 tachyzoite-infected MDBK cells in the 12-well cell culture plate were fixed with cold methanol for 1 min and stained with a Wright–Giemsa staining kit (Baso, Zhuhai, China) according to the manufacturer’s instructions. To count the number of parasites in each vacuole, at least 100 vacuoles in each group were counted.

#### 2.4.4. Detection of Cytokines

The levels of IFN-γ in mouse serum were measured using mouse IFN-γ Ready-Set-Go Kits (Thermo Scientific, San Diego, CA, USA) according to the manufacturer’s instructions.

#### 2.4.5. Neospora caninum Infection

Two weeks after the last immunization, each mouse was intraperitoneally injected with 3 × 10^7^ *N. caninum* tachyzoites. Each group included ten mice, and the survival was continuously observed for 30 days after the infection [24]. The weight of mice in each group inoculated with *N. caninum* tachyzoites was measured at 10 days after infection. In another experiment, each group also included ten mice, and the mice were sacrificed at 20 days post infection (p.i.). The brain, heart, liver, spleen, lungs and kidneys were harvested for histopathological sectioning, and the cerebral parasite burden was measured by quantitative real-time PCR (qPCR) [15].

### 2.5. Goats Immunization and Challenge

#### 2.5.1. Immunization of Goats

A total of 8 goats were divided into 4 groups (NcSRS17, NcSRS17-gIL-18, negative and positive; n = 2 per group). The goats in the negative group were not treated, and the goats in the positive group were injected with *N. caninum.* The goats in the negative and positive control groups were not immunized, whereas those in the other two groups received two inoculations at an interval of two weeks and the immunogens SRS17 and SRS17-gIL-18 at an immunization dose of 1000 µg subcutaneously, using Freund’s complete and Freund’s incomplete adjuvants at an immunization dose of 1000 µg in the first and the second immunization, respectively [25,26]. The ratio of adjuvant to immunogen was 1:1.

#### 2.5.2. Detection of Cytokines

Ten days after the last immunization, blood samples were collected from the jugular vein. The serum was collected and stored at -20 °C until use. The serum titer was detected by indirect ELISA using SRS17 and SRS17-gIL-18 as antigens as previously described [26,27]. Notably, the levels of IL-4, IL-12 and IFN-γ in goats’ serum were detected using the Goat Interleukin 12 (IL12) ELISA Kit, Goat Interleukin 4 (IL4) ELISA Kit and Goat Interleukin γ (IFN-γ) ELISA Kit (SANCHEZ, Shanghai).

#### 2.5.3. *N.*
*caninum* Infection of Goats

Each goat in the positive control, NcSRS17-immunized and NcSRS17-gIL-18-immunized groups was intravenously injected with 10^9^ *N. caninum* 15 days after collection of the serum, and the goats were sacrificed 40 days after infection [28]. The brain, heart, liver, spleen, lungs and kidneys of the goats were then harvested for histopathological sectioning, and the cerebral parasite burden was measured by qPCR [29].



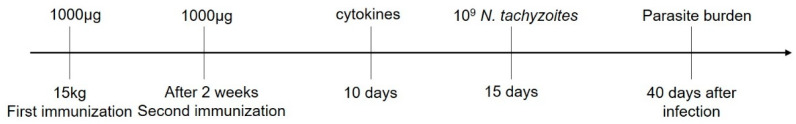



### 2.6. Statistical Analysis

All statistical analyses were performed using GraphPad Prism v5.03 (GraphPad, San Diego, CA, USA). In all cases, differences between treatment groups were considered if *p* < 0.05 marked “*****”, significant if *p* < 0.01 marked “******” and extremely significant if *p* < 0.001 marked “*******”. Furthermore, one-way analysis of variance (ANOVA) with Tukey’s multiple comparison post-test was used to analyze the differences in the antibody titers and transcriptional levels of the immune-related genes, and the differences in mortality were assessed with the chi-square test.

## 3. Results

### 3.1. In Vitro Blocking Experiment with Polyclonal Antibodies

The recombinant *N. caninum* proteins NcSRS2, NcSRS17 and NcSRS52 were purified in vitro (Figure 1A). Wright–Giemsa staining evaluated by microscopy was used to determine the number of tachyzoites in MDBK cells belonging to the NcSRS2, NcSRS17, NcSRS52 and control groups (Figure 1B). When diluted mouse serum against the recombinant protein NcSRS2, NcSRS17 or NcSRS52 was added to a tachyzoite–cell mixture, the rate of tachyzoite invasion into host cells was significantly reduced by 20% in the SRS2 group, 68% in the SRS17 group and 50% in the SRS52 group (Figure 1C). The intracellular replication of Nc-1 tachyzoites was measured by counting the number of *N. caninum* parasites in each vacuole. The results showed that NcSRS2 (*p* < 0.01), NcSRS17 (*p* < 0.001) and NcSRS52 (*p* < 0.05) could considerably inhibit the proliferation of *N. caninum*. The number of *N. caninum* parasites in infected cells belonging to group SRS17 was significantly lower than that found in the SRS2 and SRS52 groups (Figure 1D).

### 3.2. SRS17 Exerted a Better Protective Effect on Animals after N. caninum Infection

The IgG 1, IgG 2a and IgG antibody levels in mice induced by SRS2, SRS17 and SRS52 recombinant protein immunizations were detected 10 days post immunization and were significantly increased (Figure 2A). In particular, SRS17-vaccinated mice generated significantly higher antibody levels than mice vaccinated with SRS2 and SRS52 (Figure 2B). High levels of IgG1 and IFN-γ were detected in mice vaccinated with SRS2, SRS17 and SRS52 compared with those in the negative control group, suggesting the induction of a modulated Th1-type and Th2-type immune response. We also found that the IFN-γ level in the SRS17 group was higher than that in the SRS2 (*p* < 0.0001) and SRS52 (*p* < 0.0001) groups (Figure 2C).

After *N. caninum* infection, the weight loss of mice in the SRS2 and SRS52 groups was lower than those of the mice in the negative control group, but the difference was not significant; however, weight loss was significantly reduced in the SRS17 group. In addition, the weight loss of SRS17-vaccinated mice was lower than that of the mice in the SRS2 group (*p* ≤ 0.001) (Figure 3A). The qPCR results showed that the numbers of *N. caninum* tachyzoites were significantly reduced in the brains of NcSRS2-, NcSRS17- and NcSRS52-immunized mice, and the amount of *N. caninum* tachyzoites in the NcSRS17 group was lower than that in the SRS2 and SRS52 groups (Figure 3B). The survival rates of mice after challenge were found to be 16% in the positive control (*N. caninum* infection only) group, 67% in the NcSRS52 group, 83% in the NcSRS2 group and 100% in the NcSRS17 group (Figure 3C). Hematoxylin and eosin (HE) staining showed that the spleens of each experimental group showed acute oedema. The mice in the positive control group and NcSRS52 group showed sheath arteritis cell aggregation, whereas the mice in the NcSRS17 group showed mild lesions, and the spleens of mice in the negative control group did not show obvious lesions. Severe peribronchitis was found in the lung tissue of the mice in the positive control group, but no obvious pathological changes were observed in the NcSRS17 groups. Neurophagocytosis in brain tissue was detected in the positive control group, and brain oedema was found in all tissues except those of the negative control group (Figure 4). These results indicated that NcSRS17 can also be a potential vaccine candidate.

### 3.3. IL-18 Enhanced the Immunoprotective Effect of SRS2 and SRS17 against N. caninum Infection

To evaluate whether IL-18 exerted an adjuvant effect on antigens against *N. caninum*, mIL-18 was fused with NcSRS2 to generate NcSRS2-mIL-18. The results showed that the IgG 1, IgG 2a and IgG antibody levels in mice of the NcSRS2 and NcSRS2-mIL-18 groups were not changed (Figure 2B). The level of IFN-γ was significantly increased in the NcSRS2-mIL-18 group compared with the SRS2 group (*p* < 0.0001) (Figure 2C), which indicated that IL-18 played a vital role in the induction of immunity in mice. The weight loss of the mice in the NcSRS2-mIL-18 group was significantly reduced compared with that in the NcSRS2 group (*p* ≤ 0.0001) (Figure 3A). The amount of *N. caninum* tachyzoites in brain tissue in the NcSRS2-mIL-18 group was significantly lower than that in the NcSRS2 group (*p* < 0.05) (Figure 3B). The survival rates of mice after challenge were 83% in the NcSRS2 group and 100% in the NcSRS2-mIL-18 group (Figure 3C). These results suggest that IL-18 can improve the immunoprotective efficacy of antigens.

To further explore the immunoprotective role of NcSRS-17 and IL-18, goats were immunized with NcSRS-17 and NcSRS17-gIL-18. An immunoblot analysis showed that the proteins gIL-18 and NcSRS17-gIL-18 were successfully expressed and induced a strong immune reaction (Figure 5A). Ten days after the last immunization with the NcSRS-17 and NcSRS17-gIL-18 proteins, the serum titer of the NcSRS17 group was approximately 1.2 × 10^5^, whereas that of the NcSRS17-gIL-18 group was approximately 2.4 × 10^5^ (Figure 5B). The goats vaccinated with NcSRS17-gIL-18 generated significantly higher levels of IL-4 (*p* < 0.001), IL-12 (*p* < 0.01) and IFN-γ (*p* < 0.01) than those vaccinated with NcSRS17 (Figure 5C–E). Each goat was intravenously injected with 10^9^ *N. caninum*, and the goats were sacrificed at 40 days p.i. The cerebral parasite burden was measured by real-time PCR, and the number of parasites in the brain was markedly decreased in the NcSRS17 (*p* < 0.0001) and NcSRS17-gIL-18 groups (*p* < 0.0001) compared with the positive control group. The number of parasites in the brain in the NcSRS17-gIL-18 group was significantly lower than that in the NcSRS17 group (*p* < 0.01) (Figure 5F). Neurophagocytosis in brain tissue was observed in the positive control group, and brain oedema was found in all experimental groups except the negative control group. Small blood vessels were found in the liver of goats in all experimental groups, but no significant lesions were detected in the goats in the negative control group. Acute oedema was found in the spleens of goats in each experimental group, and sheath arteritis cell aggregation was also detected in the spleens of goats in the positive control group and NcSRS17 group. However, the lesions found in the goats in the NcSRS17-gIL-18 group were milder than those found in the goats in the NcSRS17 group, and the spleens of the goats in the negative control group did not show obvious lesions. Severe peribronchitis and alveolar diaphragmatic and interstitial pneumonia were found in the lung tissue of the goats in the positive control group but not in that of goats in the NcSRS17 and NcSRS17-gIL-18 groups, and the lesions in the lungs of the goats in the NcSRS17 and NcSRS17-gIL-18 groups were mild (Figure 6). These results indicated that NcSRS17 can provide protection against *N. caninum* infection in goats and that IL-18 can also increase the immunoprotective efficacy of NcSRS17.

## 4. Discussion

In the research, we focused on the SRS superfamily of *N. caninum* and aimed to select a potential vaccine candidate antigen based on our previous analyses of bioinformatics and gene specificity analysis and considering the components of the transmembrane domain and signal peptide. NcSRS17 and NcSRS52 from the SRS superfamily were considered targets, and NcSRS2 has been previously reported as a potential vaccine candidate [30], and was used as a positive control. We first evaluated and compared the immune protective effects of NcSRS2, NcSRS17 and NcSRS52 against *N. caninum* infection in mice. The results showed that mice in the NcSRS17 group exhibited less weight loss, a higher survival rate, a lower parasite load, and milder pathological changes than those in the NcSRS2 and NcSRS52 groups, indicating that NcSRS17 had good immunogenicity.

Single candidate antigens usually have limited immunogenicity, and adjuvants are needed to enhance the immunogenicity of antigens to ideally elicit both humoral and cellular immune responses. Water-in-oil adjuvants activate the innate immune system to elicit the signaling needed to initiate an adaptive immune response [30] by creating a depot effect and trapping the antigen at the site of administration, which increases the surface area available to the antigen and attracts different types of cells, including mainly APCs and macrophages [31]. In this study, water-in-oil Freund’s adjuvants exerted a positive effect on improving the immune protection induced by the antigens NcSRS2 and NcSRS17, and high IgG 1, IgG 2a and IgG antibody levels were detected, indicating humoral immune responses. The Th1-type immune response, which is defined by the production of IFN-γ and IL-12, is a characteristic of infection with intracellular microorganisms [32], such as *N. caninum*. Additionally, IFN-γ is also a critical cytokine in controlling *N. caninum* infection and promoting multiple intracellular mechanisms to kill the parasite and inhibit parasite replication.

IL-18 promotes the production of IFN-γ and exerts a synergistic effect with IL-12 [33,34]. IL-18 can also be used as an adjuvant, can regulate the response of animals to the vaccine particularly and efficiently, including recruiting immune cells to the site of vaccination and antigen presentation, promoting the proliferation, differentiation, maturation and activation of pre-T cells, T cells and B cells and activating nonspecific immune cells, including NK cells, monocytes, eosinophils and mast cells [35,36]. We previously found that IL-18 had an immunomodulatory effect [15]. Further study explored whether IL-18 could be used as an adjuvant to increase the immunoprotective role of antigen candidates in this study. We first found that IL-18 can enhance the immune effect of NcSRS2 in mice because the NcSRS2-mIL-18 group showed a higher survival rate, lower parasite burden and higher level of IFN-γ than the NcSRS2 group, indicating an immunoenhancing role of IL-18. Goats and sheep are the important intermediate hosts of *N. caninum*, and its infection in goats is globally distributed [37]. Goat IL-18 exhibits 66% homology with murine IL-18; thus, we then explored whether IL-18 plays a similar protective role in goats and tested the results in mouse experiments. We then selected NcSRS17, which we found exerted a better immune effect than NcSRS2 and NcSRS52 in mice, and fused NcSRS17 with gIL-18. Similarly, NcSRS17 immunization in goats could greatly increase the immune response and reduce the parasite load and lesions in tissues, indicating a promising antigen candidate of NcSRS17 in goats. Moreover, gIL-18 can increase the immunogenicity of NcSRS17. These results indicated that IL-18 exerts a certain immunoenhancing effect in both experimental animal mice and intermediate host goats and can be considered an adjuvant in vaccine design for neosporosis in the future.

At present, the vaccine studies for neosporosis are mainly used in mice or cattle, there are few reports on the vaccines for goats. *N. caninum* can also cause neosporosis in goats, causing huge economic losses to the industry. It is vitally important to protect goats from *N. caninum* infection. Therefore, an efficient vaccine for goats against neosporosis is urgently needed in animal husbandry. The study provides a theoretical basis for vaccine development against neosporosis, is helpful to formulate the prevention and control measures of neosporosis for goats, and, finally, ensures the healthy development of animal husbandry.

## 5. Conclusions

In summary, the recombinant proteins NcSRS2, NcSRS17 and NcSRS52 can all induce humoral and cellular immune responses, and NcSRS17 has been shown to be a promising new antigen candidate. IL-18 could be used as an adjuvant to enhance the immunogenicity of antigens.

## Figures and Tables

**Figure 1 animals-12-02952-f001:**
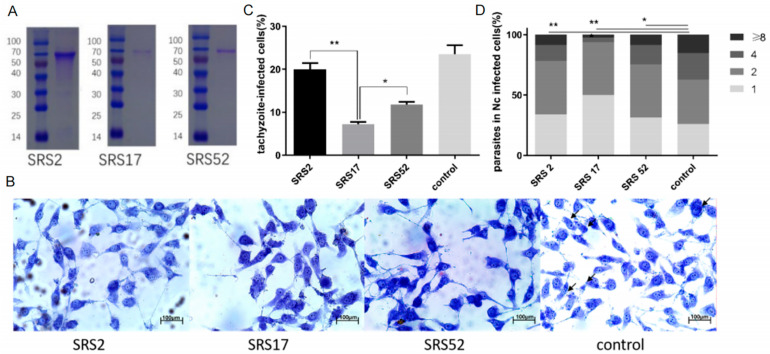
In vitro blocking experiment performed with polyclonal antibodies against the recombinant *N. caninum* proteins NcSRS2, NcSRS17 and NcSRS52. (**A**) SDS-PAGE analysis of purified recombinant proteins (NcSRS2, NcSRS17 and NcSRS52). Lane M: protein molecular weight marker; lane 1: purified recombinant NcSRS2; lane 2: purified recombinant NcSRS17; lane 3: purified recombinant NcSRS52. (**B**) Wright–Giemsa staining showing the numbers of tachyzoites in MDBK cells in the NcSRS2, NcSRS17, NcSRS52 and control groups determined by microscopy. (**C**) Inhibitory effects of polyclonal antibodies specific for the recombinant proteins NcSRS2, NcSRS17 and NcSRS52 on the invasion of *N. caninum* tachyzoites into host cells in vitro. (**D**) Inhibitory effects of polyclonal antibodies specific for the recombinant proteins NcSRS2, NcSRS17 and NcSRS52 on the proliferation of *N. caninum* tachyzoites after host cell invasion in vitro. “8” represents the number of cells containing 8 *Neospora caninum* per cell. “4” represents the number of cells containing 4 *Neospora caninum* per cell. “2” represents the number of cells containing 2 *Neospora caninum* per cell. “1” represents the number of cells containing 1 *Neospora caninum* per cell. “*” represents *p* < 0.05 and “**” represents *p* < 0.01.

**Figure 2 animals-12-02952-f002:**
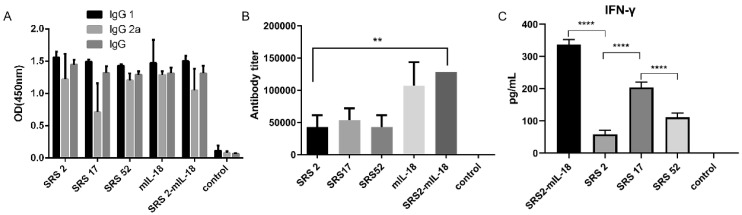
Analysis of the levels of IgG antibodies and IFN-γ in mouse serum. (**A**) Distribution of the IgG subtypes IgG1 and IgG2a in immunized mice. The levels of the IgG subtypes IgG1 and IgG2a in the serum of mice 10 days after the last immunization were analyzed by ELISA. The results are expressed as the mean OD450 ± SD values. Serum was collected one day prior to each immunization and evaluated by ELISA. (**B**) The levels of antibody titer in the mouse serum 10 days after the last immunization were analyzed by ELISA. The results are shown as the mean potency, and significantly differences are indicated by “******”. (**C**) The levels of IFN-γ in the mouse serum 10 days after the last immunization were analyzed by ELISA. Nonimmunized mice served as negative controls. “******” represents significantly different marked *p* < 0.01 and “********” represents extremely different marked *p* < 0.0001.

**Figure 3 animals-12-02952-f003:**
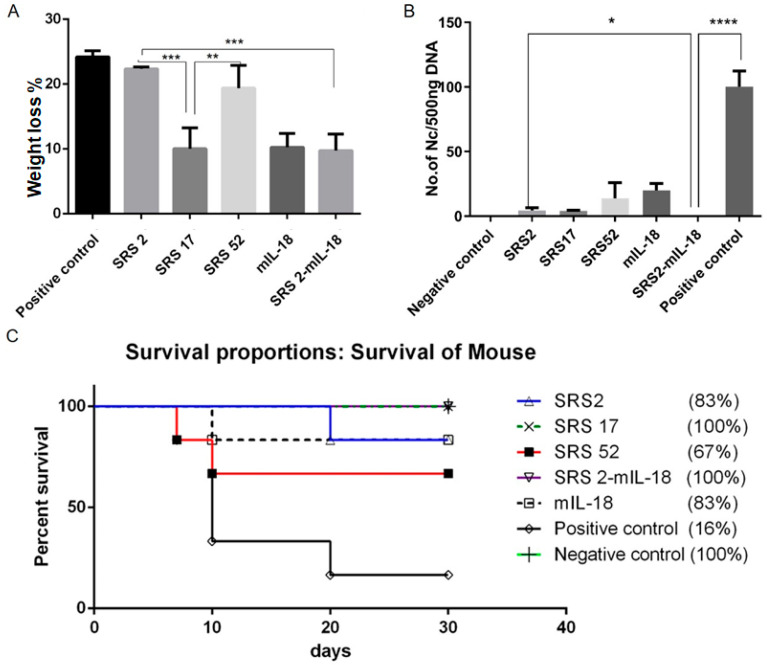
Immunoprotective effects of recombinant NcSRS2, NcSRS17, NcSRS52 and mIL18 proteins on mice after *N. caninum* infection. (**A**) Mouse weight loss results. The mice in each group were weighed before *N. caninum* infection and 10 days after *N. caninum* infection; all weights were recorded. The average weights of all the groups at the time of *N. caninum* infection were compared. (**B**) The amount of *N. caninum* in mouse brain tissue was shown by quantitative real-time PCR. (**C**) Survival curves of BALB/c mice vaccinated against *N.*
*caninum* infection. “*****” represents *p* < 0.05, “******” represents *p* < 0.01, “*******” represents *p* < 0.001, “********” represents *p* < 0.0001.

**Figure 4 animals-12-02952-f004:**
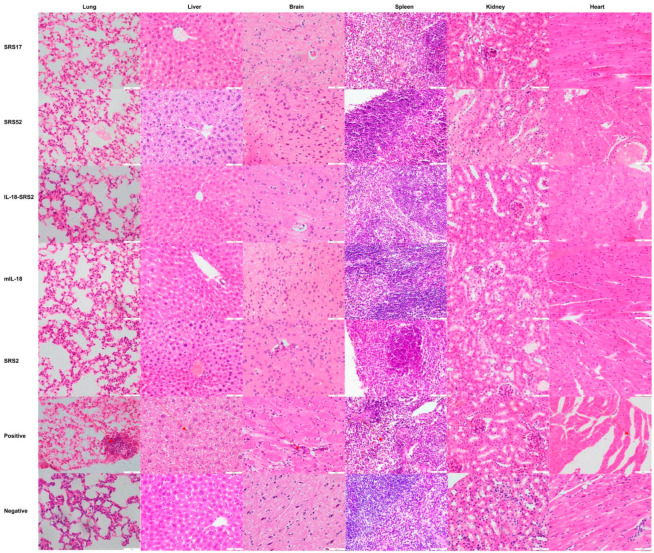
Histopathological changes in the heart, liver, spleen, lungs, kidneys and brain tissues of mice. The mouse was first immunized at 6 weeks, immunized a second time at 8 weeks and immunized a third time at 10 weeks. A histopathological analysis of heart, liver, spleen, lung, kidney and brain tissues of mice was conducted with HE staining, and observations were performed with a microscope at 20 days post infection. The brain, heart, liver, spleen, lung and kidney tissues were examined at a magnification of 200×.

**Figure 5 animals-12-02952-f005:**
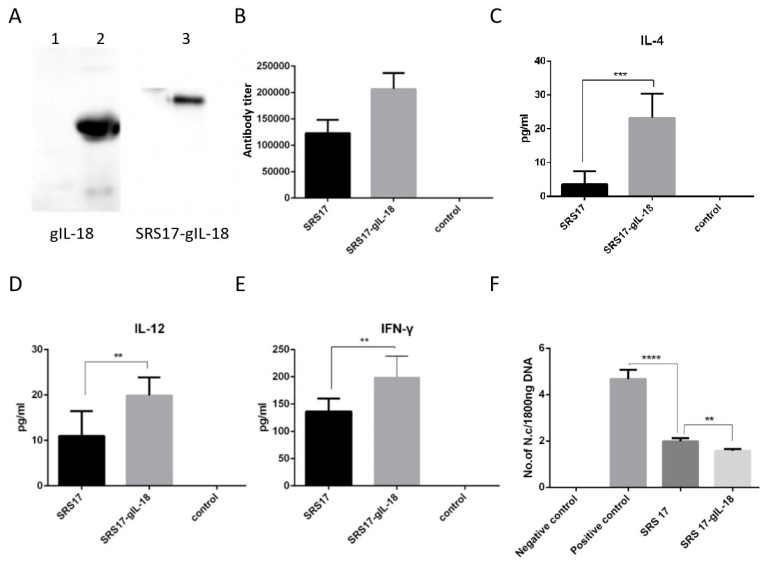
Immunoprotective effects of recombinant IL-18 and NcSRS17-gIL-18 proteins on goats after *N. caninum* infection. (**A**) Western blot analysis of the recombinant proteins gIL-18 and NcSRS17-gIL-18 with an anti-His monoclonal antibody. Lane 1: bacteria transformed with the pET-32a vector; lane 2: recombinant gIL-18; lane 3: recombinant NcSRS17-gIL-18. (**B**) Levels of IgG in the serum of goats. (**C**–**E**) Levels of IL-4, IL-12 and IFN-γ in the serum of goats. (**F**) The goats have been injected with recombinant proteins NcSRS17 and NcSRS17-gIL-18 (Goat intravenously injected with 10^9^ *N. caninum* is used the positive control and un-treated goats is used the negative control) and real-time PCR results of the brain genome in each group of goats. “******” represents *p* < 0.01, “*******” represents *p* < 0.001, “********” represents *p* < 0.0001.

**Figure 6 animals-12-02952-f006:**
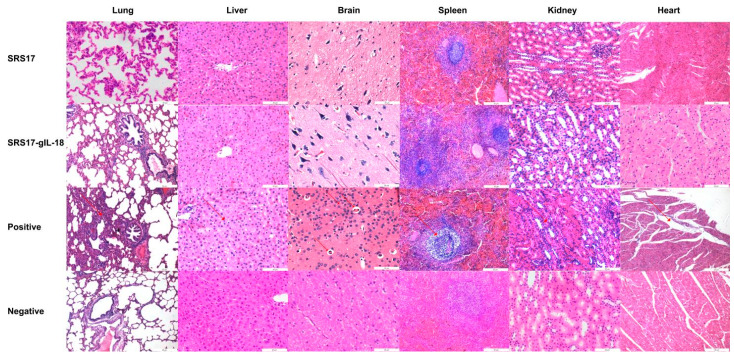
Histopathological changes in the heart, liver, spleen, lungs, kidneys and brain tissues of goats. The second immunization was carried out two weeks after first immunization in goats, and the serum was collected ten days after second immunization. Histopathological analysis of the heart, liver, spleen, lung, kidney and brain tissues of goats in the positive control, NcSRS17-immunized and NcSRS17-gIL-18-immunized groups was conducted with HE staining, and observations were conducted at a magnification of 200×.

**Table 1 animals-12-02952-t001:** Synthetic primers of the NcSRS2 (NCLIV_033250), NcSRS17 (NCLIV_068920), NcSRS52 (NCLIV_069780) or mIL-18 (NM_008360) (mature peptide) gene. It represents the digestion site marked “___” for constructed on pET-32a vector.

SRS2 (1041 bp)	Forward	5’-CCGGATATCCCGTTCAAGTCGGAAAATGAGAAGT -3’ (EcoR V)
	Reverse	5’- CCGCTCGAGGTACGCAAAGATTGCCGTTGCA -3’ (Xho I)
SRS17 (1095 bp)	Forward	5’- CCGGATATCTTTGGCAACCTGTCGTTGGATTGCG -3’ (EcoR V)
	Reverse	5’- CCGCTCGAGTTTCCCCTCGACCACAGGCAGAAA -3’ (Xho I)
SRS52 (921 bp)	Forward	5’- CCGGATATCAAGGGACGGACTAGCACTCCG -3’ (EcoR V)
	Reverse	5’- CCGCTCGAGGTACATGACCTGAACGAAGGCTACA -3’ (Xho I)
mIL-18 (471 bp)	Forward	5’-CCGGATATCAACTTTGGCCGACTTCACT -3’ (EcoR V)
	Reverse	5’- CCGCTCGAGACTTTGATGTAAGTTAGTGAGAGTG-3’ (Xho I)
SRS2-mIL-18 (1557 bp)	Forward	5’-CCGGATATCCCGTTCAAGTCGGAAAATGAGAAGT -3’ (EcoR V)
	Reverse	5’- CCGCTCGAGACTTTGATGTAAGTTAGTGAGAGTG-3’ (Xho I)
gIL-18 (471 bp)	Forward	5’- CCGGATATCTTTGGCAACCTGTCGTTGGATTGCG-3’ (EcoR V)
	Reverse	5’- CCGCTCGAGTTTCCCCTCGACCACAGGCAGAAA-3’ (Xho I)
SRS17-gIL-18 (1166 bp)	Forward	5’- CCGGATATCCCGTTCAAGTCGGAAAATG-3’ (EcoR V)
	Reverse	5’-CCGCTCGAGGTTCTGGTTTTGAACAGTGAAC-3’ (Xho I)

## Data Availability

Not applicable.

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
