# Peer review of "Evaluation of Protective Immune Responses Induced in BALB/c Mice and Goats by the Neospora caninum Surface SRS Proteins and Interleukin-18"

_animals, 2022, doi:10.3390/ani12212952_

Round 1

Reviewer 1 Report

The research entitled "Evaluation of Protective Immune Responses Induced in BALB/c 2 Mice and Goats by the Neospora caninum Surface SRS proteins 3 and Interleukin-18 " is quite novel in the development of a vaccine against N. caninum. In general the experiment is worth publication. There are some concerns which can be addressed. 

Major concerns: 

Simple summary section is missing from the document

Some English and grammatical errors are present in the manuscript that can be addressed.

Minor concerns:

Histological pictures could be magnified (showing measuring bars as reference)

Author Response

On behalf of my co-authors, we are very grateful to you for giving us an opportunity to revise our manuscript. We appreciate you very much for your comments and suggestions on our manuscript entitled "Evaluation of protective immune responses induced in BALB/c mice and goats by the Neospora caninum Surface SRS proteins and Interleukin-18" (ID: animals-1920181). We have studied reviewers′ comments carefully and tried our best to revise our manuscript according to the comments. The following are the responses and revisions we have made in response to the reviewers' questions and suggestions on an item-by-item basis. Thanks again for the hard work of the editor and reviewers!

Response to the comments of Reviewer #1

Comment No. 1: Simple summary section is missing from the document. Some English and grammatical errors are present in the manuscript that can be addressed.

Respond: Thanks for the reviewer′s kind reminder, simple summary has been added into manuscript, please see lines 18-26 in revised manuscript. English and grammatical errors have been revised by American Journal Expert company, and the certification has been provided in attachment.

Comment No. 2: Histological pictures could be revised (showing measuring bars as reference)

Respond: We thank the reviewer′s suggestion, histological pictures have been shown with measuring bars in the lower right corner of the figures.

Reviewer 2 Report

The authors present a promising strategy to develop a vaccine against N. caninum, however, I have to issues that in my opinion require further attention for the manuscript to be published.

1. They should explain much better how they expressed and purified all recombinant proteins.

2. The authors have studied much better the humoral response (protective antibodies), but they poorly studied the cellular immune response. It would be better to isolate T cells from immunized animals to see whether they could proliferate after antigen challenge. Futhermore, they should characterize those T cells.

Author Response

Dear Reviewers:

On behalf of my co-authors, we are very grateful to you for giving us an opportunity to revise our manuscript. We appreciate you very much for your comments and suggestions on our manuscript entitled "Evaluation of protective immune responses induced in BALB/c mice and goats by the Neospora caninum Surface SRS proteins and Interleukin-18" (ID: animals-1920181). We have studied reviewers′ comments carefully and tried our best to revise our manuscript according to the comments. The following are the responses and revisions we have made in response to the reviewers' questions and suggestions on an item-by-item basis. Thanks again for the hard work of the editor and reviewers!

Reviewer 3 Report

The manuscript “Evaluation of Protective Immune Responses Induced in BALB/c Mice and Goats by the Neospora caninum Surface SRS proteins and Interleukin-18” describes a study of anti-N. caninum surface proteins and their potential use as recombinant vaccines. The subject is important and interesting and fits the scope of the journal. Scientifically, the study is well designed and has potential clinical value in the development of future vaccines. However, the presentation of the study design, results and interpretation is not well constructed and difficult to follow. I suggest major English styling revision before re-submission.

General comments

-          I suggest adding a general description of the study design to the introduction or the methods section. It will make it easier for the reader to orient in the manuscript.

-          The most important clinical impact of neosporosis on farm animals is induction of abortions, which have been mostly described in cattle, but also in sheep and goats. Why were goats chosen to test the vaccine on? (and not, for example, cattle).

-          The main issue regrading this manuscript, in my opinion, is the writing. I suggest major English styling adjustments: 1) the introduction should provide a description of the importance of neosporosis, of current vaccine options and past recombinant vaccine trails, of the advantages of recombinant vaccines compared to live-attenuated vaccines, of the role of adjuvants and type of adjuvants in common use, and lastly, of the gap of knowledge this study wish to answer (and why is it important). 2) the methods section should first describe the study design, explain the choice of used proteins and adjuvant (why IL-18 and not, for example, IL-2), explain the choice of study animals (is there a reason for the choice of mouse-strain, why goats and not cattle), explain the general methodology (before getting into details) and the choice of outcome indicators (why was weight loss expected, why TNF and not other markers etc.). In the detailed description, please provide a full description of each method used, including quantities and reagents (and without skipping steps). 3) The results should be presented more clearly and fluently. Please point out to the important features in each figure and explain them clearly (for example, the histology figures are very elaborated and unclear where normal tissue is presented and where pathological findings could be found, especially in this resolution). 4) The discussion should discuss the results and not the background. What are the significant findings? What do they mean? What is their clinical importance? What is the next step towards vaccine development? Etc.

Specific comments

Title (and subtitles throughout)

-          Please put “neospora caninum” in italics.

Abstract

-          Simple summary is missing.

-          The abstract should be up to 200 words (it is currently 347).

-          Abstract, lines 21-22 – please explain what is SRS, abbreviations should be aloborated when they first appear.

Introduction

-          Throughout the text – please add a space before each citation brackets.

-          Please add a short description of what Neospora caninum is.

-          Line 45 – what is “ecological health” and what is “husbandry industry”?

-          First paragraph – Please use better citations, there are several newer publications regarding possible vaccines since 2017.

-          Line 57 – cytokines are.

-          Line 58 – please rephrase: “among their affects on immune cells, cytokines can induce…”.

-          Line 60 – interleukins.

-          Lines 62-63 – 1) Please explain what adjuvants are. 2) why specify ILs and not TNFs? Perhaps: “…cytokine adjuvants based on ILs and TNFs have been widely used…”.

-          Line 68 – Please clarify “DNA vaccine”. Most vaccines are based either on whole parasites or selected proteins.

-          Since TNF-g levels were used to determine the vaccine and adjuvant effect, please explain in the introduction the role of TNF-g, and why was it selected as a measure.

Methods

-          Please provide full description of all abbreviations when they first appear.

-          Line 79 – 1) Please start a new sentence after “respectively”. 2) Please correct to “these to media contained”.

-          Line 85 – please start a new paragraph with “Female Balb/c mice…”.

-          Line 88 – How were the goats test for NC positivity?

-          Lines 92-105 – This section was incomprehensible to me. What was the end purpose of the described process? How were the murine/caprine macrophages isolated? The tile state: plasmid construction, but the description ends at the PCR stage.

-          Table 1 should be entered after line 105.

-          Table 1 describes primers, but next to each primer there are names of restriction enzymes. Please state why in title or footnotes.

-          Lines 106-114 – Please increase the spacing between lines.

-          Line 119 and 120 – please replace “was” with “were”.

-          Line 124 – 100ug of what?

-          Line 131 – Please specify or cite the ELISA method.

-          Line 136 – Please specify the manufacturer of the specific antibodies, or the method of extraction, is these were from the immunized mice.

-          Lines 133-142 – If I understand correctly, this experiment aimed to access the neutralization efficacy of each specific antibody. If so, this should have been done prior to immunization, to suggest these proteins are good immunogens.

-          Lines 135-136 – I suspect that each antibody was done seperatly? Please clarify.

-          Lines 158-159 – please change “goat” to “goats” and “was” to “were”’ if there were two goats in each group.

-          Line 169 – Please specify or cite the ELISA method.

-          Lines 170-172 – please specify the manufacturer of the ELISA kits.

-          Line 184 – Post-hoc test.

-          Please add that the statistical comparisons were between treatment groups.

Results

-          Lines 190-192 – How can purified proteins grow in culture? Please clarify.

-          Figure 1D – Please explain what the colors and the numbers stand for in the figure legend.

-          Figure 1C and 1D – Please explain the lines and asterisks at the top of each figure in the figure legend.

-          Line 213 – please remove the word “protecting”.

-          Line 214 – What is the last IgG stand for? Total IgG? Or just IgG1 and IgG 2a? Also, please insert the word “and”, between the last two on the list.

-          Line 217 – please change “level” to “levels”.

-          Figure 2 should be placed after line 222.

-          Figure 2 legend – 1) no asterisks appear in Figure 2A, please move this notion to the end of the lagend. 2) In Figure 2B, the Y axis states “antibody titer”, while the legend states “TNF-g- levels” – please clarify.

-          Lines 223-240 are hard to follow. Please try rewriting this paragraph for more fluent reading.

-          Figure 4 – 1) please refer to the vaccination history of the mice in the legend. 2) please add a reference bar for size/length. 3) the figure is very hard to interpret. Please point out what to look at or mark important findings (by arrows, asterisks etc.).

-          Line 268 – please add a space begore the brackets.

-          Line 273 – I suggest presenting the experience in goats to a separate subsection.

-          Figure 5 legend – 1) If I understand correctly, IL-18 was not injected to goats on its own. 2) please specify what the goats in figure 5B, C, D and E were injected with, and what is the control. 3) please specify what the goats in figure 5F were injected with, and what was the target of the PCR. 4) please specify the meaning of the asterisks.

-           Figure 6 – same comments as figure 4.

Discussion

-          Line 313 – please remove the first sentence.

-          Line 314-354 – this section belongs in the introduction, as it describes the background and study design.

-          Line 358 – please change “main” (hosts) to “important”/”clinically significant”.

-          I suggest further discussion your results and their possible clinical significance to the caprine industry.

Institutional Review Board Statement

-          Please clarify if the license number provided is of the committee, or of this specific study. An ethical approval should be given specifically for each study. Also, the date of approval should be added.

Author Response

Dear Reviewers:

On behalf of my co-authors, we are very grateful to you for giving us an opportunity to revise our manuscript. We appreciate you very much for your comments and suggestions on our manuscript entitled "Evaluation of protective immune responses induced in BALB/c mice and goats by the Neospora caninum Surface SRS proteins and Interleukin-18" (ID: animals-1920181). We have studied reviewers′ comments carefully and tried our best to revise our manuscript according to the comments. The following are the responses and revisions we have made in response to the reviewers' questions and suggestions on an item-by-item basis. Please see the attachment. Thanks again for the hard work of the editor and reviewers!
